# Potato Zero-Tillage and Mulching Is Promising in Achieving Agronomic Gain in Asia

**David A. Ramírez** [1,2,*], **Cecilia Silva-Díaz** [1,3], **Johan Ninanya** [1,4], **Mariella Carbajal** [1,5], **Javier Rinza** [1], **Suresh K. Kakraliya** [6], **Marcel Gatto** [7,8] and **Jan Kreuze** [1]

1 International Potato Center (CIP), Headquarters, P.O. Box 1558, Lima 15024, Peru; silvadiaz.cc@gmail.com (C.S.-D.); j.ninanya@cgiar.org (J.N.); m.carbajal@cgiar.org (M.C.); j.rinza@cgiar.org (J.R.); j.kreuze@cgiar.org (J.K.)
2 Water Resources Doctoral Program, Universidad Nacional Agraria La Molina (UNALM), Lima 15024, Peru
3 Agricultural Sciences and Resource Management in the Tropics and Subtropics (ARTS), University of Bonn, 53113 Bonn, Germany
4 Department of Meteorological Engineering and Climate Risk Management, Universidad Nacional Agraria La Molina (UNALM), Lima 15024, Peru
5 Biological and Agricultural Engineering, North Carolina State University, Campus Box 7625, Raleigh, NC 27695, USA
6 International Potato Center (CIP), Patna 800014, India; suresh.kakraliya@cgiar.org
7 International Potato Center (CIP), Hanoi 100000, Vietnam; m.gatto@cgiar.org
8 School of Economics and Finance, University of the Witwatersrand, Braamfontein, Johannesburg 2000, South Africa
* Correspondence: d.ramirez@cgiar.org; Tel.: +51-993-913-578

**Abstract:** Rice-based systems have recently been recognized as the most critical plant source of C emissions worldwide. Globally, rice production is highest in Asia. Actions to introduce sustainable intensification practices into existing rice lands or diversifying with lower C-emitting crops such as potatoes will be crucial to mitigate climate change. The objective of this study is to analyze the effect of potato cultivation under zero/minimum-tillage and/or organic mulching (with emphasis on rice-straw) (PZTM) on key performance indicators that are crucial to achieving agronomic gains in Asia. Forty-nine studies were selected and systematically reviewed to address the study objective. Studies reveal a consensus of increase in yield, profitability, nutrient-use efficiency, and water productivity, promoted by the significant soil moisture conservation in PZTM. There is inconsistent evidence that zero-tillage benefits weed control, but its effectiveness is enhanced by mulching. Even if soil organic matter is increased (+13–33%) and zero-tillage is the main factor driving the reduction in C footprint, no values of kg $CO_2$ eqha$^{-1}$ have been reported in PZTM to date. Only a small fraction (~2%) of the rice-cultivated areas (RCA) is intensified with potato cultivation. That way, scaling-up PZTM among rice farmers has large potential (~24% RCA) to increase the sustainable intensification of rice-based systems in Asia.

**Keywords:** conservation agriculture; *Solanum tuberosum*; sustainable intensification; rice-based system; soil health; C footprint

## 1. Introduction

The increase in the productivity of existing lands, while reducing environmental impacts, is a core challenge in Sustainable Intensification (SI). It requires conservation agriculture, integrated pest management, agroforestry, etc., [1,2], and appropriate policies [3]. The dramatic increase in per capita agricultural production [2], acute rural poverty and environmental degradation [4], and the rise in extreme climatic events related to water risk, threaten future agricultural production [5], especially in Asia, where SI will be crucial [6]. In Asia (especially in South and Southeast Asia, China, and Mongolia), the GHG emissions caused by plant-based food are the highest worldwide, and rice cultivation appears to be the main driver of emissions through $CH_4$ release [7]. Rice is the principal nourishment

for over 60% of the biosphere's inhabitants [8], and rice-based systems occupy the most important agricultural lands in some of the vastest countries of Asia [9]. Environmental problems such as carbon emission from rice-residue burning, the overexploitation of underground water table, land degradation, and productivity reductions in rice-based agricultural systems have been reported in Asia [6,10]. On the other hand, there have been significant efforts to adopt SI in these systems [11–13], with substantial results for soil recarbonization [14].

Potato has been considered an essential alternative for intensifying rice-based systems in fallow areas between rice cultivation or rice and other crops (wheat) [15–18]. In rice-dominant countries, intensification with potato could diversify diets and create additional incomes [19]. For that purpose, genetic gains through breeding activities to achieve early-maturity potatoes (70–90 days) that are well-adapted to local conditions have gained importance to ensure appropriate yield during the short window between rice and subsequent crops [20,21]. However, it has recently been observed that more than productivity alone is needed to improve genetic gain [22]. Thus, agronomic gain is a current, more inclusive concept related to reducing yield gaps through practices that aim to improve productivity, resource use efficiencies, and soil health. Agronomic gain considers different environments, is socially inclusive, and can easily be framed as a key performance indicator (KPIs) [22]. Based on recent findings in cereals-based cropping systems in South Asia [13], we hypothesized that the SI of rice-based systems with potato could benefit KPIs and other indicators if certain components of conservation agriculture, such as a reduction in mechanical soil damage (through zero or minimum tillage) and the incorporation of organic matter in the soil are considered. The objective of this study is to present a review of scientific evidence of zero-tillage (and synonymous or related terms, see Table 1) and organic mulching (with an emphasis on rice-straw; see details in Table 1) effects in several KPIs (related to productivity, resources-use efficiency, and soil health), C footprint, and weed control for growing potatoes in rice-based systems in Asia.

**Table 1.** Search equations performed at each database used in this review. In gray, the codes refer to document types in BASE database search query. n—number of articles.

| Database | Search Equation | n |
| --- | --- | --- |
| Scopus (Naïve search) | TITLE-ABS-KEY("potato" AND ("zero till*" OR "no till*" OR "*straw mulch*" OR "organic mulch*") OR "rice-wheat system*") | 174 |
| Scopus (Litsearchr) | TITLE-ABS-KEY ( ( "mulch*" OR "organic mulch*" OR "tillage with mulching" OR "paddy straw*" OR "straw mulch*" OR "no-till*" OR "conservation tillage" OR "direct drilling" OR "direct seeding" OR "zero till*" OR "zone till*" OR "conventional tillage" ) AND ( "potato*" OR "solanum tuberosum*" ) ) | 345 |
| Google Scholar | potato* + (mulch* OR "organic mulch*" OR "tillage with mulching" OR "paddy straw*" OR "straw mulch*" OR "no-till*" OR "zero-till*" OR "zone till" OR "conservation OR "conventional tillage" OR "direct drilling" OR "direct seeding") +"asia" | 220 |
| BASE | ((conservation sustainable intensification) agriculture) (productivity efficiency) AND (((zero no minium) till) mulch rice) AND potato AND asia -europe doctype:( 122 13 14 15 16 17 18* 19 1A ) | 227 |

## 2. Materials and Methods

### 2.1. Systematic Review

Identification, screening, and selection processes were followed in this systematic review (Figure 1, INPLASY registration number 202260072). Regarding the identification process, the important literature regarding rice-based systems SI and agronomic gain [17,22,23] was used to extract keywords for the search query in a naïve search performed in the Scopus database. Subsequently, the "litsearchr" script (R package v1.0.0; [24]) was used in a semi-automated search term selection for systematic reviews. This package uses a slightly simplified version of the Rapid Automatic Keyword Extraction (RAKE) method that includes text-mining and keyword co-occurrence networks to identify potential terms from abstracts and titles [25]. Scopus, which is considered the most extensive database for indexed and peer-reviewed research, including 100% PubMed and with twice as many indexed journals as Web of Science [26], was used to search the scientific literature.

Google Scholar and BASE databases were also used to include the grey literature for an extensive literature review. With a more refined search equation from "litsearchr", and new terms that could have been overlooked in the naïve search, different combinations with Boolean operators were created according to the database format (Table 1). Under the screening process (Figure 1), after many attempts, the extracted documents were subjected to a first screening applied in the titles, where those studies made in Asia that included terms "zero-till*" and/or "mulch" (and related ones, see Table 1), based on rice-wheat systems (and related crops), and available in the English language were selected after removing duplicates. Then, the potentially relevant records from the databases mentioned were joined together in one list for a second screening, which was used in the abstracts and, when required, the complete text. During the screening process, 213 documents were identified (Figure 1). Only those focused on potato crops with the full text available in English were kept for the selection process. The references from this new list were revised in the full-text form to group them according to KPI (yield, profitability, water-nutrient efficiency, and soil health), C footprint, and weed control using Google forms, and these groups were used for the results of this review. In the selection process (Figure 1), 49 documents were selected for a full review after the final exclusion according to the eligibility criteria.

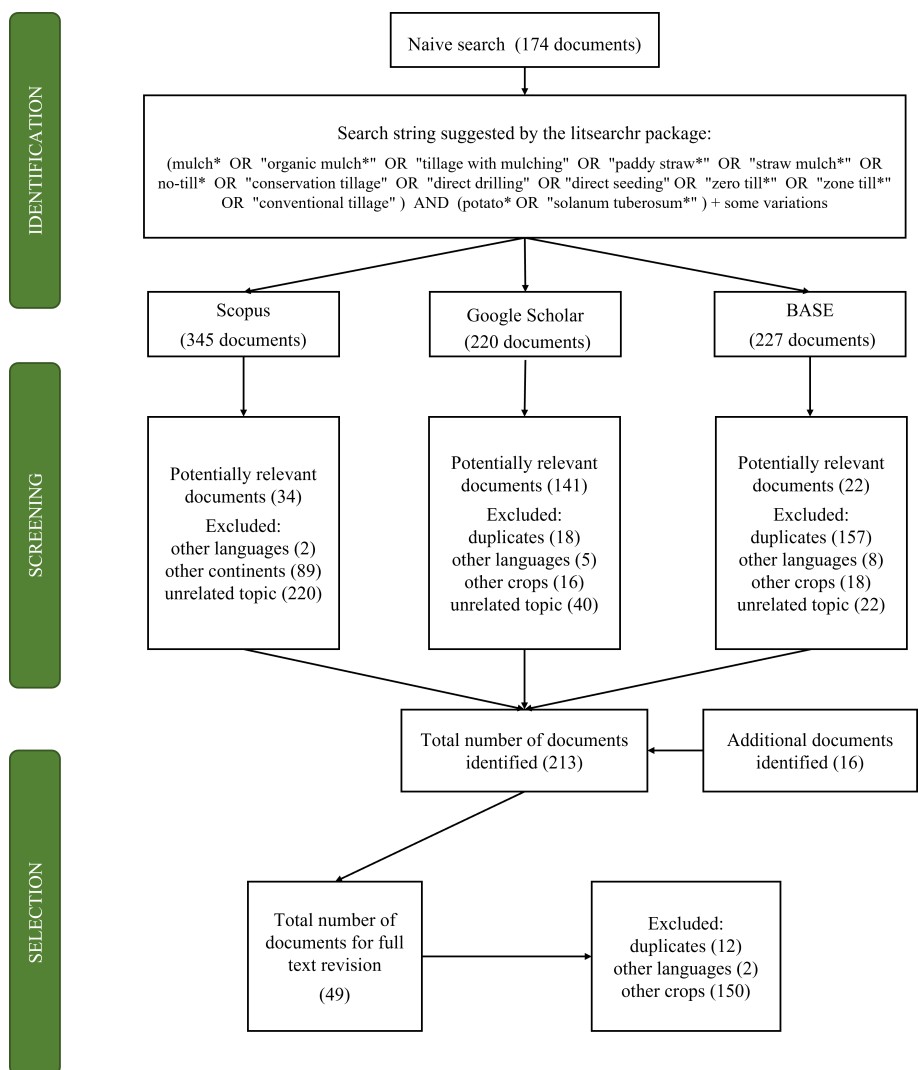

**Figure 1.** Flow diagram of the search methodology during 20–25 March 2022.

### 2.2. Analysis of the Literature

A qualitative classification of the evidence level of all indicators analyzed (KPI, C footprint and weed control) was performed according to Locatelli et al. [27]'s methodology.

Thus, four categories (Locatelli et al. [27]'s categories) based on the level of coincidence among studies (LC) and the number of studies (NS) were defined as follows: (i) "Consensus" (medium or high LC and high NS), (ii) "Probable" (medium or high LC and few NS), (iii) "Controversy" (low LC and high NS), and (iv) "Knowledge gap" (low LC and few NS). Medium or high LC was considered when most of the studies coincided in their results (same increase/decrease trend in KPIs indicators), whereas 10% (or less) of the studies were considered as few NS [27,28].

*2.3. Potential Rice-Potato Area Calculation in Asia*

The MODIS Land Surface Temperature/Emissivity Daily L3 Global 1 km (MOD11A1) version 6.1 product was used within the Google Earth Engine Cloud Platform to determine the potential potato production areas. The minimum and maximum daily average air temperature (2010–2020) were estimated from daytime and nighttime MODIS land surface temperature, respectively, according to Zhu et al. [29]'s procedure. Thus, areas where maximum temperatures were less than 35 °C and minimum temperatures ranged between 2 °C and 21 °C during at least 100 days during the November–March sowing window [30] were used to map the potential potato production areas. In addition, maps of the estimated harvested areas of rice and potatoes were downloaded from the Spatial Production Allocation Model (SPAM 2010 v2.0 Global Data). This software is a global gridded mathematical model developed by the International Food Policy Research Institute [31] and validated in many crops, especially rice [32,33]. ArcGIS 10.5 [34] was used to determine maps of the current rice–potato and rice–non-potato rotation areas (areas under 100 ha were not considered due to their low representativeness in the 10,000-ha pixel coverage). Finally, the maps (from SPAM) of rice and non-potato rotation areas were masked with the map of potential potato production areas to estimate the actual rice production zones that could be potentially be rotated with potatoes in Asian countries. The raster calculator and zonal statistics toolbox of ArcGIS 10.5 [34] was used to achieve this.

## 3. Results and Discussions

*3.1. Context and Antecedents for Moving Forward to PZTM in Asian Rice-Systems*

Agronomic gain, as a KPI related to productivity, requires a combination of improved agronomic practices that allow for high and stable yields and economic profitability [22] to be obtained. Growing an additional crop (e.g., potato) within a rice-based system produces more food on the same land area at the smallholder level, which is in the line with SI principles [35]. It has been reported that mulching has a significantly positive effect on potato crops improving seedling emergence, plant height, numbers of stems, and tuber yield [36]. In addition, zero-tillage practices promote early seeding because seed tubers are planted immediately after rice harvest [37–39]. According to those authors, conventional tillage used many resources (e.g., time, energy, money, and labour) and required a significant investment of labor in rice systems. At least 80% of the studies on zero-tillage with organic mulching were conducted in different regions of India, Bangladesh [36,40], China [41], and Indonesia [42] between 2000 and 2018, including one from the 1970s [43]. The area harvested under rice–potato was estimated to be around 3.02 Mha in Asia (Figure 2), based on the modeling performed in this study, around 33.6 Mha of the rice area (~24%), which could additionally be intensified with short-duration crops such as potato during the winter season [9,17,44].

The principal conservation technique evaluated in this review are organic mulch from paddy straw applied to potato crops, followed by its combination with zero-tillage. Plastic mulch and other organic materials for mulching, such as wheat, pine-needle, reed *Thypa* sp., and the grass *Pennisetum lyphoides* [41], were also mentioned in some studies. The management of rice straw is a significant challenge for farmers in these regions because it is considered a poor feed for animals due to its high silica content [45]. Farmers burn crop residues because they can not leave them on the field due to their long decay period and they can spread diseases from the last/previous paddy season [15], as well as their

having short sowing period window that does not allow farmers to manually clear the fields [45,46]. South Asian countries, including India, Pakistan, Sri Lanka, Bangladesh, Afghanistan, Nepal, and Bhutan, have a crop residue production potential of 1172 Mt, of which the highest is in India (912 Mt), specially in the rice–wheat system. About 372 Mt residues are surplus [47].The burning of crop residues is a particular issue in Punjab, but it is also increasing in the Eastern States of Indo-Gangetic Plains [48]. Crop-residue burning is a potential source of emission of $CO_2$, as well as pollutants such as carbon monoxide CO, particulate matter, and toxic polycyclic aromatic hydrocarbons [49,50]. Due to stricter regulations, farmers are looking for non-burning alternatives, and mulching is a convenient, sustainable practice in regions where straw resources are locally available [51]. Together with zero-tillage, this can help to reduce burning and, thus, air pollution [45].

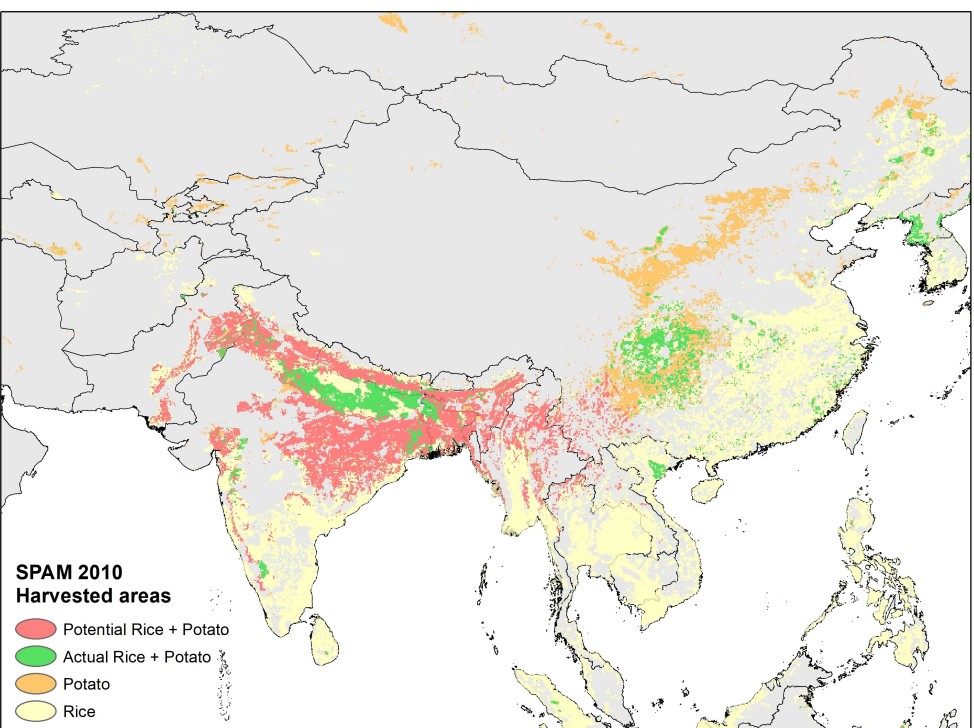

**Figure 2.** Potato- and rice-harvested areas estimated from SPAM 2010 [31] for Asian countries.

### 3.2. Productivity KPI: Yield and Profitability

Most of the reviewed studies analyzed the relationship between conservation techniques and the yield (71% of the total) and profitability (53% of the total). They all reported that both methods enhance crop yield and profitability, evidencing a "consensus" in Locatelli et al. [27]'s the productivity category KPI (Figure 3). In most cases, tuber yields under mulch increase by 20% or more compared to non-mulched production [37,52–54] (see details in Table 2). Other studies have achieved increases of up to 100% [55] and 173% [42] (Figure 4). The effect of organic mulch on potato yield remained similar when it was assessed on different soil types, with a greater increase compared to yield without a mulch cover [56]. In addition, mulch can mitigate water and heat stresses by increasing soil moisture and reducing soil temperature, respectively, promoting a higher yield [57] (Figure 4). The combination of mulch and a closer distance (between 30 and 60 cm between plants and rows, respectively) was more appropriate when comparing different plant spacings because the lower soil temperature promoted a better accumulation of carbohydrates in the tubers [42]. The reduced soil temperature is considered the main factor that improves potato plant development [43]. This could be due to the low heat transmissivity, which resulted in a more significant fraction of solar radiation being absorbed at the top of the mulch layer [53]. Numerous studies have applied different evenly spread mulch quantities

(5–9 t ha$^{-1}$, [36,38,53,54,56,58,59]). Sarangi et al. [39] reported a 20-cm layer of rice straw mulching (12 t ha$^{-1}$) and foliar application of fertilizers as the optimum treatment to guarantee the appropriate quantity and quality of the tuber yield. These authors emphasized that an optimum thickness of paddy straw mulch should be maintained to avoid the tubers greening. Finally, to promote significant efficiency in crop production, Ref. [60] recommended using crop residue mulch and minimum tillage, which would significantly increase the fiber content of the tubers [39].

**Table 2.** Summary of the evidence of zero-tillage and/or organic mulch effects in potatoes on Saito et al. [22]'s key performance indicators and C footprint and weed control. The percentages of increase (↑) or decrease (↓) referred to the values obtained under control condition (i.e., under conventional practices without mulching and/or reducing/zero-tillage). REF = Reference.

| Technique | Combined with | Location | Key Performance Indicators | | | | | Weed Control and/or C Footprint | REF |
|---|---|---|---|---|---|---|---|---|---|
| | | | Productivity | | Resource Use Efficiency | | Soil Health | | |
| | | | Yield | Profitability | WP * | NUE | SOC, SAL | | |
| M | irrigation levels and soil textures | Ludhiana, India | ↑ 22–31% | 13,200 INR ha$^{-1}$ of return | ↓ 90–100 mm IW ↓ T° ↑ θ | 90 kg N ha$^{-1}$ saved | | | [56] |
| M | irrigation schedules | Madhya Pradesh, India | ↑ 10% | ↑ 4% NR | ↑ 10% WUE | ↑ P and K uptake | | | [52] |
| M | other mulched crops | Assam, India | ↑ 21% | ↑ 33% NR | | | | | [61] |
| M | irrigation schedules | Odisha, India | ↑ 24–42% | | | ↑ P and K uptake | | | [53] |
| M | plant spacing levels | East Java, Indonesia | ↑ 173% | | ↓ T° | | | | [42] |
| M | plastic mulch | China | ↑ 12–17% | | ↑ 8% | | | | [41] |
| M | herbicide (atrazine) | Ludhiana, India | ↑ 46% | | | | | up to 99% of weeds controlled | [58] |
| M | herbicides (metribuzin, clodinafop) | Ludhiana, India | ↑ 100–135% | only mulch: 397 USD ha$^{-1}$ NR, with herbicides: 62–513 USD ha$^{-1}$ NR | | | | 68–95% of weeds controlled | [55] |
| M | K concentration levels | Bangladesh | | ↑ 16–53% NR | ↑ 28–107% | | | | [36] |
| M | other mulched crops, many potato varieties | Assam, India | ↑ 20% | ↑ 18–29% NR | | | | | [54] |
| M | Different planting dates | Gerua, Kamrup, India | ↑ 24% | | | | | ↓ 16% incidence of common scab | [37] |
| M | irrigation methods and schedules | Ludhiana, India | ↑ 41% | ↑ 811% NR | ↑ 42%, ↑ drainage, ↓ T°, ↑ θ | | | | [59] |
| M | irrigation levels | Odisha, India | ↑ 5% | ↑ 24% NR | ↑ 5% WUE | | | | [62] |
| M | N concentration levels | Ludhiana, India | | | ↑ T° | | ↑ 33% SOC | | [63] |
| ZTM | early sowing of potato | Coastal region of India | ↑ 6% | ↑ 96% NR ↑ 60% BCR | ↓ 40% IW | | | | [38] |
| ZTM | various tillage practices | Ludhiana, India | ↑ 12% | ↑ 16% BCR | | | | | [45] |
| ZTM | foliar nutrients | West Bengal, India | ↑ 24–93% | ↓ 27% CC ↑ 147–498% NR | ↓ 31–38% IW | | ↑ 13% SOC, ↓ 40% soil salinity | | [39] |
| ZTM | fertilizers and other crops | Ludhiana, India | ↑ 4% | ↑ 24% BCR ↑ 16% NR | | | | | [60] |
| ZTM | other crop systems | New Delhi, India | ↑ 7% | ↑ 13% NR | ↑ 28% | | | | [64] |
| ZTM | different organic mulch materials | Bangladesh | | | ↑ 4–8% θ | | ↓ 4% soil salinity | | [40] |

BCR: benefit–cost ration, CC: cost of cultivation, INR: Indian rupees, IW: irrigated water, M: paddy straw mulch, NR: net return, RPFI: ridge planting furrow irrigation, MT: minimum tillage, SOC: soil organic carbon, T°: soil temperature, USD: United States Dollar, WUE: water use efficiency (weight of harvested tubers per evapotranspiration), WP: water productivity (weight of harvested tubers per quantity of irrigated water), ZTM: zero-tillage and paddy straw mulching, θ: soil moisture. * Into WP category it is also considered associated indicators (WUE, IW) and drivers of soil water conservation (T°, θ, drainage).

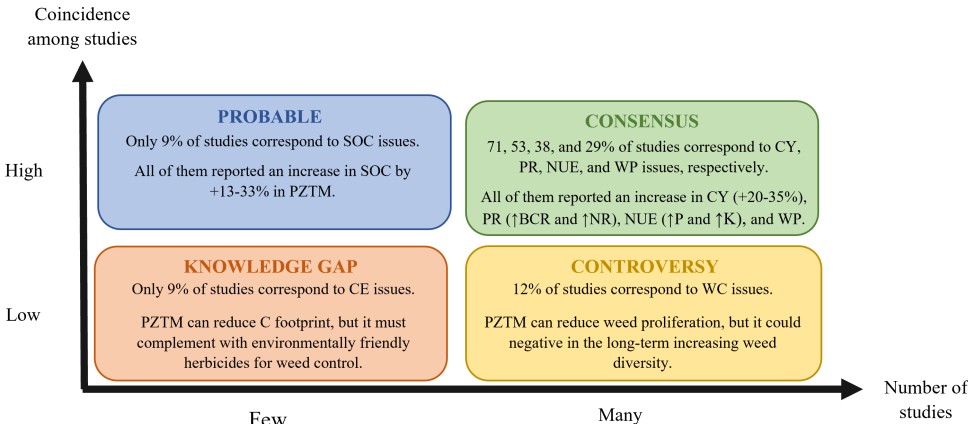

**Figure 3.** Classification of the evidence level of potato cultivation under zero-tillage and/or organic mulch (PZTM) effects on Saito et al. [22]'s Key Performance Indicator (KPI) as a function of coincidence among studies and number of studies [27]. Four or fewer studies are considered as "few". Analyzed KPIs included: Productivity (CY—crop yield and PR—profitability), resources efficiency (WP—water productivity and NUE—nutrient-use efficiency), soil health (SOC—soil organic carbon), CE—C footprint, and WC—weed control. BCR—benefit–cost ratio. NR—net return. P—phosphorus content. K—potassium content.

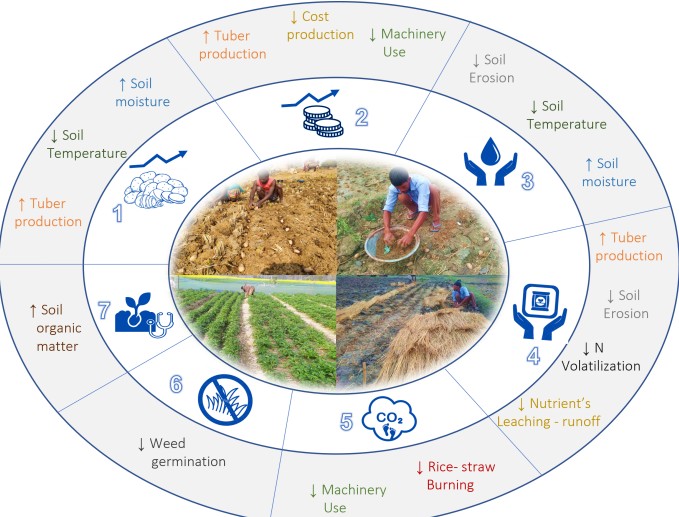

**Figure 4.** The main processes triggered by zero/minimum-tillage and/or mulching under potato cultivation reported by the literature that improve crop yield (1), profitability (2), nutrient-use efficiency (3), water productivity (4), C footprint (5), weed control (6) and soil organic carbon (soil health) (7).

The most common potato variety cultivated in India (mainly in Karnataka and West Bengal [21]) was "Kufri Jyoti," followed by "Kufri Chandarmukhi" and "Kufri Pukharj". Compared to other winter crops cultivated in the fallow season, such as tomato, pea, or toria, the potato has the highest net return (11.5 rupees ha$^{-1}$) under conventional agronomical practices. This increased by 46% if paddy straw mulching was used [54]. On average, mulching increases the benefit–cost ratio (BCR; i.e., revenues per rupee invested) by +10% over no-mulching practice (BCR = 1.55) [62] (see details in Table 2). Notwithstanding, if combined with an optimum irrigation level, BCR and net return can be increased by +30 and +10%, respectively [52,62]. In addition, Brar et al. [59] reported an average increase of gross return, the net return, and BCR by +10, +71, and +70%, respectively, combining straw mulch with different planting methods and irrigation (ridge-planting furrow irrigation, bed-planting furrow irrigation, ridge-planting drip irrigation, and bed-planting drip

irrigation). Rice-straw mulching is the most appropriate material for reaching higher yield and increasing the economic benefits, obtaining a BCR value of 2.61 compared to water hyacinth or sawdust [36]. Although the cost of crop residue for mulching application is higher than that needed for tillage operations, studies have shown that the combination of zero/minimum tillage with mulching has better gross/net return and monetary efficiency than conventional tillage practice [64]. In the specific case of the rice–wheat system, applying zero-tillage reduces the cost of cultivation through savings in seeds, tillage, fertilizers, transplanting, and irrigation. Still, with the higher cost of weed control, the benefit of this practice is expressed in the higher yield in most cases with lower production costs [65] (Figure 4). If residues (water and fertilizers) were still present, then fewer inputs would be needed in the following season [15]. However, although in situ crop residue management can save money in the long-term, mulching has not been widely adopted by farmers because additional fuel and energy costs and machinery are required to produce mulch from, for instance, rice straw [66]. The combine harvester cuts the crop while the machine moves through the field, threshes the crop, and scatters the straw throughout the field, but this activity can be perofrmed manually if there is enough available labor [44]. These authors have proposed a model where the harmonization and synchronization of farming operations of numerous Indian small farmers could increase their incomes and minimize the scale disadvantages, in addition to a reduced C footprint due to fuel emissions and the burning of residues [67].

### 3.3. Resource Efficiency KPI: Water Productivity and Nutrient-Use Efficiency

Rice uses the principal portion of the total volume of water used for worldwide crop production [68], and the subsequent largest water consumer is wheat [8]. Potato is an irrigated crop in the Indo-Gangetic plains of India, which requires about 8–10 irrigations during the crop season [56]. Some of the reviewed studies (29%) combined mulch cover with irrigation schedule treatments, from which mulched crops recorded an increment of 10% in water productivity (WP) over non-mulched crops [52,53]. In other cases, this technique significantly reduced the irrigation requirement to half of the irrigation pulses [38]. The reduction in water evaporation promoted by a low solar energy incidence under organic mulching conditions leads to increased soil moisture conservation in potatoes [15,53,59] (Figure 4). However, the data are very variable; for instance, Li et al. [41] reported an increase in WP by 7.7% at air temperatures ranging from 15 to 20 °C without significant effects over 20 °C (see details in Table 2), and Goel et al. [69] registered a 13.5% increment of WP with paddy straw mulch. Li et al. [41] also stated that straw mulching significantly increased potato WP by 8.3% in areas with low water input, but had no significant effect in areas with high water input. When combining zero-tillage and rice-straw mulching, about 200 mm of irrigation (compared with conventional tillage) water can be saved, thereby reducing the water footprint [39].

Some other studies (38%) have considered the relationship between conservation techniques and soil nutrition, all of which generated a "consensus" in Locatelli et al. [27]'s category (Figure 3) to increase nutrient-use efficiency in PZTM compared to conventional tillage. Thus, there is evidence [52,53] of a significant increment in P and K but not in N uptake by the soil when paddy straw mulch is added to potatoes. Therefore, straw mulch may reduce nutrient loss, especially the volatilization of N fertilizer, and increase nitrogen use efficiency by reducing N losses [67] (Figure 4). Bijay-Singh et al. [70] note that 80–85% of the K absorbed by rice and wheat remains in their straw, and its incorporation as residue also enhances the level of soil organic and inorganic P, reducing its sorption and thereby allowing it to substitute about 13 kg ha$^{-1}$year$^{-1}$ inorganic P [71]. Combining the application of K (125 kg ha$^{-1}$) with rice straw mulch, Pulok et al. [36] reported the highest number of tubers per hill and tuber yield (28.9 t ha$^{-1}$). Additionally, organic mulching can maintain a higher soil and plant water status over non-mulched soil with only half of the amount of N (60 vs. 120 kg N ha$^{-1}$, respectively), resulting in a water-saving of 40 mm without affecting leaf area index and yield [72] (Figure 4).

### 3.4. Soil Health KPI

Regarding the increase in soil organic content (SOC) by PZTM, this was considered "probable" in the Locatelli et al. [27]'s category due to the few studies that were carried out (9%, Figure 3). Thus, Bhagat et al. [63] reported an increase of 33% compared to non-mulched soils (Table 2). Similarly, Sarangi et al. [39] observed a 12.8% increment in soil organic carbon (SOC) by combining rice–straw mulching and zero-tillage (Figure 4). Thus, PZTM could enhance soil quality and carbon sequestration by preventing soil erosion, leaching, and runoff of nutrients (enhancing nutrient use-efficiency, Figure 4), boosting soil microbial diversity and associated enzymatic activity and restricting weed infestation [73–75]. Soil treated with crop residues contained 5–10 times more aerobic bacteria and 1.5–11 times more fungi than soil from which residues were either burned or removed [66]. However, it should be noted that, to obtain a significant improvement in soil health through minimum tillage, a certain set of 3–5 years is required [76]. Sarangi et al. [38] and Akkas et al. [40] reported a soil salinity reduction under PZTM treatments in coastal areas (Table 2); they suggest that thiscould be caused by the reduction in soil water evaporation promoted by soil temperature reduction and moisture retention.

### 3.5. C Footprint

The C footprint measures the emission of gasses that contribute to heating the planet in $CO_2$ equivalents per unit of time or product [8]. It is crucial to identify the C footprint of a crop; thus, appropriate strategies can be devised to reduce C footprint and make agriculture more sustainable [77]. Only a few studies (9% of the total) studied C footprint, which suggests that there is a substantial knowledge gap regarding the quantitative effects of minimum/zero-tillage and/or organic mulching on C footprint in rice–potato systems (Figure 3). In the SI of agricultural systems, strategies to mitigate climate change should include agronomical practices that reduce C footprint (by maximizing fertilizer use efficiency), enhance carbon sequestration, and avoid $CO_2$ emissions from fossil fuels [8]. Inherently, agricultural practices (tillage, fertilizer use, crop burning, land use, among others) contribute to $CO_2$ production; however, crop residue mulching and minimum/zero-tillage enhance soil carbon sequestration [78] (Figure 4). Lal [79] reported that conventional tillage produces $\sim$35 kg C kg$^{-1}$, whereas zero and minimum tillage released 6–8 kg C kg$^{-1}$. A carbon sequestration rate of 0.2–0.5 t C ha$^{-1}$ can be reached by combining best-management practices such as minimum tillage and fertilizer use and crop residue mulching [78]. In potatoes, zero-tillage paddy straw mulching can increase SOC from 0.39 to 0.44%, enhancing carbon sequestration [39]. Tanveer et al. [80] reported that zero-tillage produces the lowest C footprint compared to all the other tillage practices. The C footprint is higher (+20–68%) in the double rice agricultural system rotated with potatoes than that rotated with other crops, such as rape or Chinese milk vetch [81]; still, it can be reduced by minimum/zero-tillage with residue crop mulch practices (Figure 4).

### 3.6. Weed Control

In potatoes, the proliferation of weeds reduces the number and size of tubers [58] by depleting the resources available to the potato crop and interfering with its growth by releasing allelochemicals and harboring harmful insects and pathogens that reduce yield [55]. There is a "controversy" if weed control is improved in PZTM in the reviewed literature (Figure 3). On the one hand, the combination of mulching with different concentrations of herbicides (such as atrazine, clodinafop plus metribuzin) avoided up to 70% of weed species in potatoes [55,58] (see Table 2). Moreover, Sarangi et al. [39] state that rice-straw mulching can eliminate weed germination, so there is no need for the intercultural operations required in conventional practices. In wheat cultivation, zero-tillage has been found to be more effective in reducing weed germination and growth than conventional tillage [82]. Conversely, Bhatt [76] reported that zero-tillage plots have a significantly higher weed population because tillage places the seed deeper in the soil, where reduced moisture and nutrient availability limit weed germination (Figure 4). Likewise, Murphy et al. [83]

considered that the continuous suppression of tillage could increase weed diversity and the proliferation of novel weeds. In addition, the decomposition of residues, such as mulch, may promote weed emergence and growth by improving soil fertility; therefore, Chaudhary et al. [67] argued that zero-tillage and mulching compel growers are dependent on herbicides to manage weeds. Finally, as conventional weed control usually requires frequent herbicide applications, and sometimes additional tractors [58], avoiding these activities might reduce the C footprint that they would cause.

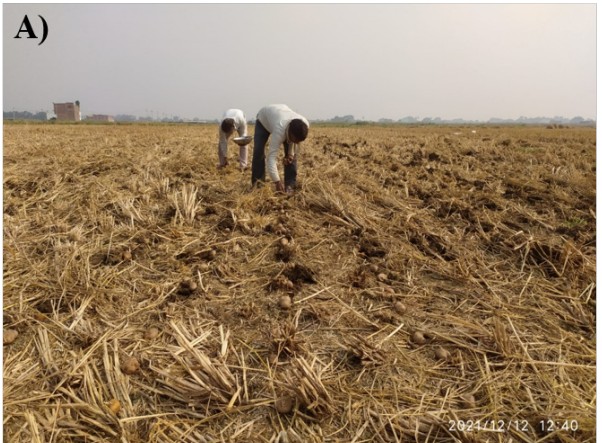 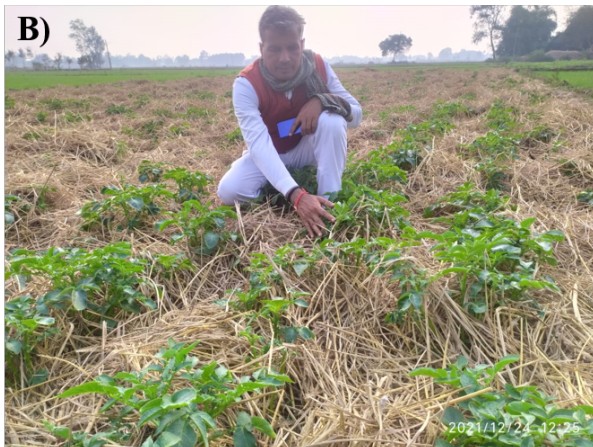

**Figure 5.** Potato cultivation under zero-tillage and rice-straw mulching (PZTM) on-farm demonstration trials in Patna district, Bihar-India. After rice harvesting, potatoes seeds are placed in soil (**A**) and covered by manure and rice-straw mulch. Around 30 days after plating, emerged plants overpass the mulch (**B**). Photo credit: Suresh K. Kakraliya.

### 3.7. Additional Considerations before Moving Forward to PZTM Technology

The dissemination of potato diseases through degenerated seeds is one of the most significant constraints on potato production [84] and is considered a major problem for potato cultivation in Asia [44]. The use of quality seeds guarantees appropriate yield through planting material with a low incidence of diseases. Potato apical rooted cuttings (i.e., rooted transplants produced from sterilized tissue culture plantlets) have emerged as a low-cost seed multiplication alternative, in contrast to other technologies (such as aeroponic) used in Asia [85]. The use of early-maturing varieties (i.e, harvest crop 75–90 days after planting) allows for farmers to take advantage of the temporal window between rice and wheat [20,21]. However, the adoption of these new early-maturing potato varieties, such as Kufri Khyati, Lady Rosetta and Kufri Pukhraj, adapted for India [21], needs to be considered in future analyses. Since earliness is a critical trait in the fight against climate change in many important potato-producing countries in Asia [21], studies on the adoption of those promising varieties are necessary. Ramírez et al. [86] highlights the potential of advanced potato clones to tolerate extreme saline soil conditions, and the breeding efforts have led to the release of varieties in Bangladesh. The development of varieties from promising clones with high salt tolerance (such as CIP 396311.1 and CIP 301029.18) and ex-post/ante adoption studies will be an important research topic to better understand the agronomic, socio-economic, and environmental effects of PZTM technology.

### 4. Conclusions

There is a need to promote, facilitate and expand PZTM in Asia with academy and research institutions, NGOs, the private sector, and governments. These efforts need to consider four critical points to achieve agronomic gain. Once access to clean seed of appropriate genotypes (precocious and salt-tolerant) is achieved, smallholder farmers will need simple protocols to implement PZTM technology. This aspect is critical for non-experienced farmers in potato cultivation. The use of "community videos", consisting of audiovisual

resources created by farmers [87], could be powerful tools for disseminating PZTM. These tools have been propagated in Asia with critical success [88,89], allowing a clear and direct way for knowledge transfer due to those videos being created in local languages. The third critical point is to guarantee women's participation and support in PZTM. This technique has been referred as "women-friendly potato production technique" [44]. Recent reports in India have highlighted the impact of this technology on women's engagement, promoted by the reduction in labor (no ploughing, planting or digging is needed [15]; see Figure 5), and the participation of women can be improved through an increase in visual information, language, and using women's images in training materials [90]. Finally, to be well prepared for future carbon markets and compensations, it is necessary to quantify the reduction in C footprint that is achieved by growing PZTM compared to conventional tillage. Easy, friendly and open-access calculators such as Cool Farm Tool (CFT; [91]) or CCAFS-MOT [92] are available for C footprint inspections, with successful application in potato [93,94].

There is confident evidence of improved yield, profitability, water productivity, and nutrient-use efficiency, as well as a probable increase in SOC levels, which clearly suggests that PZTM is a highly promising technology for smallholder farmers. As such, PZTM can significantly contribute to the agro-ecological transformation many countries are currently undergoing. If PZTM is combined with available early-maturing varieties, traditional cereal-based systems can be sustainably intensified with potatoes. There is a huge opportunity (~33.6 Mha or ~24%), given that only a small percentage of rice area (~2%) in Asia is currently used to grow potatoes, providing an opportunity to increase SI through PZTM in Asia. However, the effects of PZTM on C footprint is an under-researched topic, whereas limited "controversial" evidence was found regarding weed control. In addition, most of the analyzed studies in this review used "controlled" experimental fields as the study method. Research is needed that explores PZTM under farmer conditions and its socio-economic effects on rural livelihood outcomes. Studies that examine the determinants of adoptions and pathways to increase adoption rates and scaling of the technology are equally important. Finally, quantifying the extent to which C emissions are reduced in PZTM will be crucial for future compensation schemes for small farmers that promote SI agriculture in rice-based systems in Asia.

**Author Contributions:** Conceptualization, D.A.R., C.S.-D. and J.N.; methodology, C.S.-D. and J.N.; software, J.N. and M.C.; formal analysis, C.S.-D., J.N. and M.C.; investigation, D.A.R., C.S.-D., J.N. and J.R.; resources, M.G. and J.K.; data curation, J.N. and C.S.-D.; writing—original draft preparation, D.A.R., C.S.-D. and J.N.; writing—review and editing, J.R., S.K.K., M.G. and J.K.; visualization, D.A.R., J.N. and M.C.; supervision, M.G.; project administration, M.G. and J.K.; funding acquisition, M.G. and J.K. All authors have read and agreed to the published version of the manuscript.

**Funding:** This research was undertaken as a part of, and funded by The Deutsche Gesellschaft für Internationale Zusammenarbeit (GIZ) through the fund for the promotion of innovation in agriculture (i4Ag): "Potato production through zero-tillage with straw mulch: an innovative technology for sustainable intensification and diversification of rice-based systems to improve livelihoods of small-scale farmers in Asia" (Agreement N°: 81275993), and the CGIAR Research Program on Roots, Tubers and Bananas: RTB_PO2.5.5.1 and is supported by CGIAR Trust Fund contributors https://www.cgiar.org/funders/, accessed on 1 May 2022.

**Institutional Review Board Statement:** Not applicable.

**Informed Consent Statement:** Not applicable.

**Data Availability Statement:** Not applicable.

**Acknowledgments:** The authors want to thank Suresh K. Kakraliya for his generous support in establishing potato cultivation under zero-tillage and rice straw mulching of demonstration trial in farmers field of this project in Bihar. The pictures shown in this review are evidence of this critical work.

**Conflicts of Interest:** The authors declare no conflict of interest.

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
