# Peer review of "Potato Zero-Tillage and Mulching Is Promising in Achieving Agronomic Gain in Asia"

_agronomy, doi:10.3390/agronomy12071494_

Round 1

Reviewer 1 Report

With respect to the Figure 1, the Flow diagram and eligibility criteria to reach only 46 papers should be better or further explained in the text.

Subsection on Potential rice-potato area calculation in Asia merits stight development by authors and add references of previous uses of models and estimations.

Figure 3 still requires additional explanation from authors to be simple for readers. Authors are asked to explain litte more the evidence level of potato cultivation.

Lines 123 - 126 something is missing in relation to previous paragraph.

Table 2 is very interesting and original work and hence authors are asked to develop further.

Figure 4 is an extremely important part of this research review paper and merit to be better presented and explained.

Subsection on future directions should be more focused on the insights from the research paper. or may be combine it with conclusions.

Author Response

Responses to the comments of the reviewers of the manuscript ID: agronomy-1741447 "Potato zero-tillage and mulching is promising in achieving agronomic gain in Asia"

We would like to thank you for the thorough and very constructive review, which we found very helpful. We agree with the recommendations and have changed the corresponding parts of the manuscript accordingly. All the reviewers’ comments were addressed and responded in the same order, as follows:

REPLIES TO REVIEWER # 1’S COMMENT

Review # 1 – Comment 1

With respect to the Figure 1, the Flow diagram and eligibility criteria to reach only 46 papers should be better or further explained in the text.

Our response

We appreciate your suggestion. Considering your suggestion, the three processes (Identification, Screening and Selection) in our new flow diagram (Fig. 1) have been referred in the text as follow:

Page 2, lines 56-59:

Identification, screening, and selection processes were followed in this systematic review (Fig. 1). Regarding the identification process, important literature about regarding rice-based systems SI and agronomic gain [17,22,23] was used for extracting keywords for the search query in a naïve search performed in the Scopus database.

Page 2-3, lines 70-74:

Under the screening process (Fig. 1), after many attempts, the extracted documents were subjected to a first screening applied in the titles where those studies made in Asia that included terms “zero-till*” and/or “mulch” (and related ones, see Table 1), based on rice-wheat systems (and related crops), and available in English language were selected after removing duplicates.

Page 3, lines 74-82:

Then, the potentially relevant records from the databases mentioned were joined together in one list for a second screening used in the abstracts and, when required, the complete text. In the screening process 213 documents were identified (Fig. 1). Only those focused on potato crops with full text available in English were kept for the selection process. The references from this new list were revised in full text to group them according to KPI (yield, profitability, water-nutrient efficiency and soil health), C footprint, and weed control by using Google forms, which were used for the results of this review. In the screening   selection process (Fig. 1), 213 49 documents were identified, and were selected for a full review after the final exclusion according to the eligibility criteria.

Review # 1 – Comment 2

Subsection on Potential rice-potato area calculation in Asia merits stight development by authors and add references of previous uses of models and estimations.

Our response

We appreciate your comment. Attending your suggestion, we have included a short description of the methodology, and two references regarding previous use of this global gridded mathematical model were added as follows (Page X, lines X-X):

Page 4, lines 94-111:

“The MODIS Land Surface Temperature/Emissivity Daily L3 Global 1km (MOD11A1) version 6.1 product was used within the Google Earth Engine Cloud Platform to determine the potential potato production areas. The minimum and maximum daily average air temperature (2010-2020) were estimated from daytime and nighttime MODIS land surface temperature, respectively, according to Zhu et al. [28]’s procedure. Thus, those areas where maximum temperatures were less than 35â—¦C and minimum temperatures ranged between 2â—¦C and 21â—¦C during at least 100 days during the November-March sowing window [30] were used to map the potential potato production areas. In addition, maps of the estimated harvested areas of rice and potatoes were downloaded from the Spatial Production Allocation Model (SPAM 2010 v2.0 Global Data). This software is a global gridded mathematical model developed by the International Food Policy Research Institute [31] and validated in many crops, especially rice [32, 33]). ArcGIS 10.5 [34] was used to determine maps of the current rice-potato and rice-non-potato rotation areas (areas under 100 ha were not considered due to their low representativeness in the 10000-ha pixel coverage). Finally, the maps (from SPAM) of rice and non-potato rotation areas were masked with the map of previously calculated of potential areas potato production areas to estimate the actual rice production zones that could potentially be rotated with potatoes in Asian countries. For this aim, using the raster calculator and zonal statistics toolbox of ArcGIS 10.5 [34] was used.”

[32] Yu, Q., You, L., Wood-Sichra, U., Ru, Y., Joglekar, A. K., Fritz, S., ... & Yang, P. (2020). A cultivated planet in 2010–Part 2: the global gridded agricultural-production maps. Earth System Science Data, 12(4), 3545-3572. https://doi.org/10.5194/essd-12-3545-2020

[33] Yang, M., Wang, G., Lazin, R., Shen, X., & Anagnostou, E. (2021). Impact of planting time soil moisture on cereal crop yield in the Upper Blue Nile Basin: A novel insight towards agricultural water management. Agricultural Water Management, 243, 106430. https://doi.org/10.1016/j.agwat.2020.106430

Review # 1 – Comment 3

Figure 3 still requires additional explanation from authors to be simple for readers. Authors are asked to explain litte more the evidence level of potato cultivation.

Our response

We appreciate this suggestion. Thus, a more detailed explanation of Figure 3 was added as follows:

Page 3-4, Lines 84-92

A qualitative classification of the evidence level of all indicators analyzed (KPI, C footprint, and weed control) was performed according to Locatelli et al. [27]’s methodology. Thus, four categories (Locatelli et al. [27]’s categories) based on the level of coincidence among studies (LC) and the number of studies (NS) were defined as follows: i) "Consensus" (medium or high LC and high NS), ii) "Probable" (medium or high LC and low few NS), iii) "Controversy" (low LC and high NS), and iv) "Knowledge gap" (low LC and low few NS). Medium or high LC was considered when most of the studies coincided in their results (same increase/decrease trend in KPIs indicators), whereas 10% (or less) of the studies were considered as few NS [27, 28]

Page 6, Lines 150-152

They all reported that both methods enhance crop yield and profitability, evidencing a “consensus” in Locatelli et al. [27]’s category for productivity KPI (Fig. 3).

Page 8, Lines 221-223

Some other studies (38%) have considered the relationship between conservation techniques and soil nutrition, all of which generated a “consensus” in Locatelli et al. [27]’s category (Fig. 3) for increasing nutrient-use efficiency in PZTM compared to conventional tillage.

Page 8, Lines 237-239

Regarding the increase in soil organic content (SOC) by PZTM, it was considered "probable" in the Locatelli et al. [27]'s category due to the few studies (9%, Fig. 3). Thus, Bhagat et al. [61] report an increase of 33% compared to non-mulched soils (Table 2).

[28] Bonnesoeur, V., Locatelli, B., & Ochoa-Tocachi, B. (2019). Impactos de la Forestación en el Agua y los Suelos de los Andes: ¿Qué sabemos?. CGIAR Infraestructura Natural para la Seguridad Hídrica, Forest Trends, Lima, Peru.

https://agritrop.cirad.fr/591482/1/Bonnesoeur%202019%20Impacto%20de%20la%20Forestacion%20en%20el%20Agua%20y%20Suelos.pdf

Review # 1 – Comment 4

Lines 123 - 126 something is missing in relation to previous paragraph.

Our response

We agree. We consider that these lines provide a piece of detailed information that is not necessary under this context. To guarantee internal coherence, we deleted this phrase and the next one (regarding Bangladesh) (see page 5, line 125).

Review # 1 – Comment 5

Table 2 is very interesting and original work and hence authors are asked to develop further.

Our response

We appreciate your comment. Table 2 provides specific information regarding used techniques, locations, ranges of values of KPIs, and references. This Table has been referred in the text in our first version to complement the discussion (see page 7, lines 154; page 7, lines 178; page 8, lines 215, page 8, lines 249); however, we realized that some data was missing in the Table. This missing data has been included in the new Table2 version (see the new Table 2 version in the Overleaf, where the added data was remarked in yellow).

The citation of Table 2 was added in the following pages – lines:

Page 9, lines 279-281

On the one hand, the combination of mulching with different concentrations of herbicides (like atrazine, clodinafop plus metribuzin) avoided up to 70% of weed species in potatoes [56, 57] (see Table 2).

Review # 1 – Comment 6

Figure 4 is an extremely important part of this research review paper and merit to be better presented and explained.

Our response

Attending your suggestion this figure has been modified to improve its presentation (see new Fig. 4 , page X). The Figure caption has been changed as follows:

Page 9

Figure 4. The Mmain processes triggered by zero/minimum-tillage and/or mulching under potato cultivation reported by the literature that improves crop yield (1), profitability (2), nutrient-use efficiency (3), water productivity (4), C footprint (5), weed control (6) and soil organic carbon (7).   some key performance indicators to achieve agronomic gain related to productivity, resource use efficiency, and soil health (sensu Saito et al. [22]). C footprint reduction and weed control were also included as a part of this review.

This figure summarizes (in a graphical way) the mechanisms reported in our review that improve Key Performance Indicators, weed control, and C footprint. Thus, once the text remarked on the scientific evidence, this figure was cited to reinforce the explanation graphically. Even though this figure was referred to in the text in our first version (see current page 7, line 158; page 8, line 211-212; page 8, line 225-227; page 8, line 242-243, page 9, line 271-273 and page 10, line 284-286), in this new manuscript version, we have checked if it was necessary to cite this figure in additional lines to attend to your suggestion. The citation of Fig. 4 was added in the following pages – lines:

Productivity KPI (Section 3.2):

Page 7, line 154

Other studies have achieved increases up to 100% [57] and 173% [42] (Fig. 4).

Page 7, lines 189-192

In the specific case of the rice-wheat system, applying zero-tillage reduces the cost of cultivation through savings in seeds, tillage, fertilizers, transplanting, and irrigation. Still, with a higher cost of weed control, the benefit of this practice is expressed in a higher yield in most of cases with lower production costs [62] (Fig. 4).

Resource Efficiency KPI (Section 3.3):

Page 8, Lines 232-235

Also, organic mulching can maintain a higher soil and plant water status over non-mulched soil with only half of the amount of N (60 vs. 120 kg N ha−1 respectively), resulting in a water saving of 40 mm without affecting leaf area index and yield [69] (Fig. 4).

Soil Health KPI (Section 3.4):

Page 8, Lines 240-241

Similarly, Sarangi et al. [39] observed a 12.8% increment in soil organic carbon (SOC) combining rice-straw mulching and zero-tillage (Fig. 4).

Carbon Footprint (Section 3.5):

Page 9, Lines 261-264

Inherently, agricultural practices (tillage, fertilizer use, crop burning, land use, among others) contribute to CO2 production; however, crop residue mulching and minimum/zero-tillage enhance soil carbon sequestration [76] (Fig. 4).

Review # 1 – Comment 7

Subsection on future directions should be more focused on the insights from the research paper. or may be combine it with conclusions.

Our response

We agree. Following your recommendation, we have moved this sub-section into the Conclusion section (page 11, line 314-350). To focus only on our research insights, we have removed specific technologies regarding soil C assessment using reflectometers (page 11. lines 333-340 of the previous version).

Reviewer 2 Report

Dear Authors,

The article is interesting and discusses important environmental issues. The manuscript was prepared in the form of a scientific review, not a research article (for the editorial decision) - the Authors themselves indicate: 2. Materials and Methods-; 2.1. Systematic Review + 2.2. Analysis of the literature.

Detailed comments below:

1) Table 1 - why was the Web of Science database omitted?

2) Chapter No. 3.7. Additional considerations before moving forward to PZTM technology - is it possible to indicate specific potato varieties (with different early stages)? Take into account that the division of plants only depending on the variety's earliness does not take into account their phenological fit to the cultivation area.

3) Chapter 4. Conclusions - this chapter is summarized (I suggest changing to "Summary"). Additionally, emphasize the applicability aspect.

4) Lines 54-61, in my opinion the content is redundant.

Author Response

Responses to the comments of the reviewers of the manuscript ID: agronomy-1741447 "Potato zero-tillage and mulching is promising in achieving agronomic gain in Asia"

We would like to thank you for the thorough and very constructive review, which we found very helpful. We agree with the recommendations and have changed the corresponding parts of the manuscript accordingly. All the reviewers’ comments were addressed and responded in the same order, as follows:

REPLIES TO REVIEWER # 2’S COMMENT

Review # 2 – Comment 1

The article is interesting and discusses important environmental issues. The manuscript was prepared in the form of a scientific review, not a research article (for the editorial decision) - the Authors themselves indicate: 2. Materials and Methods-; 2.1. Systematic Review + 2.2. Analysis of the literature.

Detailed comments below:

1) Table 1 - why was the Web of Science database omitted?

Our response

Thank you for this comment. It was because the Scopus database is a larger curated database that covers almost all scientific journals, books, conference proceedings, etc., indexed in the Web of Science database. Thus, for example, Singh et al. (2021), comparing the journal coverage of Web of Science and Scopus databases, found that ~99% (~34% of all journals indexed in Scopus) of the journals indexed in Web of Science are also indexed in Scopus.

Singh, V. K., Singh, P., Karmakar, M., Leta, J., & Mayr, P. (2021). The journal coverage of Web of Science, Scopus and Dimensions: A comparative analysis. Scientometrics, 126(6), 5113-5142. https://doi.org/10.1007/s11192-021-03948-5

Review # 2 – Comment 2

2) Chapter No. 3.7. Additional considerations before moving forward to PZTM technology - is it possible to indicate specific potato varieties (with different early stages)? Take into account that the division of plants only depending on the variety's earliness does not take into account their phenological fit to the cultivation area

Our response

We agree with your suggestion. This information has been added as follows:

Page 10, Lines 302-308

“The use of early-maturing varieties (i.e, harvest crop in 75-90 days after planting) allows farmers to take advantage of the temporal window between rice-wheat [20, 21]. However, the adopting on of these new potato early-maturing varieties, such as Kufri Khyati, Lady Rosetta and Kufri Pukhraj adapted for India [21], needs to be considered in future analyses. Considering that Since earliness is a critical trait to in the fight against climate change in many important potato-producing countries in Asia [21], studies of the adoption the prediction of those promising varieties adoption are necessary is promising.”

Review # 2 – Comment 3

3) Chapter 4. Conclusions - this chapter is summarized (I suggest changing to "Summary"). Additionally, emphasize the applicability aspect

Our response

Thanks for this comment. We have moved the future research actions sub-section into the Conclusion section to include the applicability aspect of our research. With this modification, we comply with your requirement. Please see the new Conclusions section on Page 11, line 315-350 of the new manuscript version.

Review # 2 – Comment 4

4) Lines 54-61, in my opinion the content is redundant.

Our response

We are fully agreed, these lines have been deleted (see Page 2, line 53).
